# MEND: Meta dEmonstratioN Distillation for Efficient and Effective In-Context Learning

**Yichuan Li**[1][*], **Xiyao Ma**[2], **Sixing Lu**[2], **Kyumin Lee**[1], **Xiaohu Liu**[2], **Chenlei Guo**[2]
[1]Worcester Polytechnic Institute, [2]Amazon Alexa AI
{yli29,kmlee}@wpi.edu
{maxiya,cynthilu,derecliu,guochenl}@amazon.com

## Abstract

Large Language models (LLMs) have demonstrated impressive in-context learning (ICL) capabilities, where a LLM makes predictions for a given test input together with a few input-output pairs (demonstrations). Nevertheless, the inclusion of demonstrations leads to a quadratic increase in the computational overhead of the self-attention mechanism. Existing solutions attempt to distill lengthy demonstrations into compact vectors. However, they often require task-specific retraining or compromise LLM's in-context learning performance. To mitigate these challenges, we present **M**eta d**E**monstratio**N D**istillation (MEND), where a language model learns to distill any lengthy demonstrations into vectors without retraining for a new downstream task. We exploit the knowledge distillation to enhance alignment between MEND and LLM, achieving both efficiency and effectiveness simultaneously. MEND is endowed with the meta-knowledge of distilling demonstrations through a two-stage training process, which includes meta-distillation pretraining and fine-tuning. Comprehensive evaluations across seven diverse ICL task partitions using decoder-only (GPT-2) and encoder-decoder (T5) attest to MEND's prowess. It not only matches but often outperforms the Vanilla ICL as well as other state-of-the-art distillation models, while significantly reducing the computational demands. This innovation promises enhanced scalability and efficiency for the practical deployment of large language models [1].

## 1 Introduction

Large language models (LLMs) have demonstrated exceptional power in in-context learning (Kaplan et al., 2020; Brown et al., 2020; Dong et al., 2023; Min et al., 2022a). They can rely on a limited number of input-output pairs, often termed demonstrations, to generate outputs for a given test input, without parameter updates. However, a significant bottleneck arises: incorporating demonstrations exacerbates input length for LLMs. This is concerning, especially considering the self-attention mechanism inherent in these models, which imposes time and memory complexities that scale quadratically with input length.

Attempts to mitigate this challenge typically focus on trimming the context length by distilling extensive demonstrations into concise vectors as shown in Fig. 1. These vectors are then used to prompt the LLM to generate outputs (Phang et al., 2023; Ivison et al., 2022; Mu et al., 2023; Lester et al., 2021). Distillation approaches, however, differ across methodologies. For instance, methods such as prompt tuning (Lester et al., 2021; Wang et al., 2023) produce vectors through gradient descent. Nonetheless, these approaches necessitate specific retraining for different demonstrations. In contrast, the introduction of hypernetworks (Ha et al., 2016) offers a solution that reduces the reliance on gradient descent for any given demonstrations. Methods like Hypertuning (Phang et al., 2023) and HINT(Ivison et al., 2022) employ conditional language modeling (CLM) objectives to finetune a language model based distillation model, distilling demonstrations into vectors. Yet, when benchmarked against the Vanilla ICL method—where LLMs are prompted directly with the unaltered demonstration text—the performance exhibits discernible degradations using these distilled

---

[*]This work was mainly done during Yichuan's internship at Amazon.

[1]The code is avaliable at https://github.com/bigheiniu/MEND.

vectors. This trend remains consistent, even when distillation models are co-trained with the `LLM` in ICL data (Ivison et al., 2022). Given that these language model based distillation models inherently possess in-context learning capabilities and can generate meaningful representations, the remaining question is how to optimize them to generate demonstration distillation that rival or even surpass the efficacy of `Vanilla ICL`. Achieving this would pave the way for enhancing ICL efficiency without compromising its efficacy.

During pretraining, `LLMs` usually learn using detailed word data. But at demonstration distillation scenario, they have to work with a simplified version of this data – distilled vectors. It's like studying with a full textbook but taking the test with only a summary. We think it's really important to make sure that the LLM can understand and use these summaries just as well as the full textbook. This helps the LLM perform better when it's actually being used for ICL. To address this, we introduce the **M**eta d**E**monstration **N D**istillation (`MEND`). Our approach realigns the distillation model, `MEND` and LLM through knowledge distillation (Hinton et al., 2015; Snell et al., 2022). Here, the LLM, when prompted solely with the distilled vectors (acting as the *student*), is conditioned to emulate the behavior it would exhibit when exposed to the full demonstrations (assuming the role of the *teacher*). To achieve this, we minimize the Kullback–Leibler (KL) divergence between teacher and student models' word distributions. Importantly, during this optimization process, we backpropagate the gradients from the `LLM` to `MEND`, while ensuring that the LLM remains frozen throughout. The training paradigm for `MEND` is twofold: meta-distillation pretraining on standard text pretraining data (e.g. C4 (Raffel et al., 2019)), followed by finetuning on ICL tasks. This two-stage training equips `MEND` with the meta-knowledge for distilling demonstrations, allowing it to generalize effectively across unseen demonstrations without sacrificing performance.

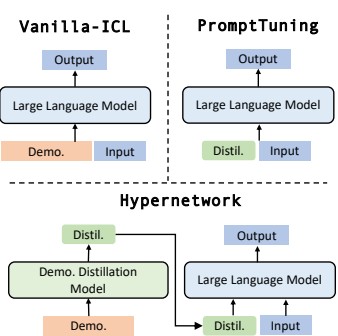

Figure 1: `Vanilla ICL` method utilizes the concatenation of demonstrations and test input to generate the output. In contrast, `PromptTuning` and `HyperNetworks` employ distilled vectors in place of the full demonstrations. The length of these distilled vectors is significantly shorter than that of the demonstrations, contributing to a more compact and efficient in-context learning for LLM.

To demonstrate the feasibility of `MEND`, we apply it to a variety of LLM architectures, including both decoder-only (e.g., `GPT-2`(Brown et al., 2020)) and encoder-decoder configurations (e.g., `T5 (Raffel et al., 2019)`). In our experiments on the `MetaICL` dataset (Min et al., 2022a), encompassing 142 unique NLP tasks divided across seven partitions, `MEND` consistently meets or exceeds the performance of `Vanilla ICL`, notably outperforming where traditional hypernetwork approaches falter. Across the range of language models we investigated, our distillation strategy results in a substantial reduction of up to $75\%$ in FLOPs and accelerates inference by up to $33\%$. Beyond standard evaluations, we embarked on an in-depth diagnostic analysis where we tweaked the distillation ratio and added intentional disturbances to the demonstrations. In these scenarios, `MEND` proved resilient to the disruptions and consistently outpaced standard `Vanilla ICL` methods.

Summarizing our work, our contributions are threefold: (1) The introduction of `MEND`, an innovative technique aimed at enhancing the `LLM`'s in-context learning efficiency without compromising the performance; (2) An exploration into the benefits of knowledge distillation for aligning the demonstration distillation model with LLM; (3) Comprehensive quantitative and qualitative examinations that highlight the robustness and effectiveness of `MEND`.

## 2 PROBLEM DEFINITION

Let $\mathcal{D} = \{(x_i, y_i)\}_{i=1}^K$ be a demonstration set, where $x_i$ and $y_i$ denote the input and output tokens respectively, and $K$ is the number of input-output pairs or demonstrations. Let $D$ denote the concatenation of demonstration set that is $D = \texttt{concat}(x_1, y_1, \cdots x_K, y_K)^2$. In in-context learning (ICL), given $D$, and test input $x$, the large language model (LLM) will compute the conditional

---

[2]In the following sections we will use concatenated demonstrations and context interchangeably.

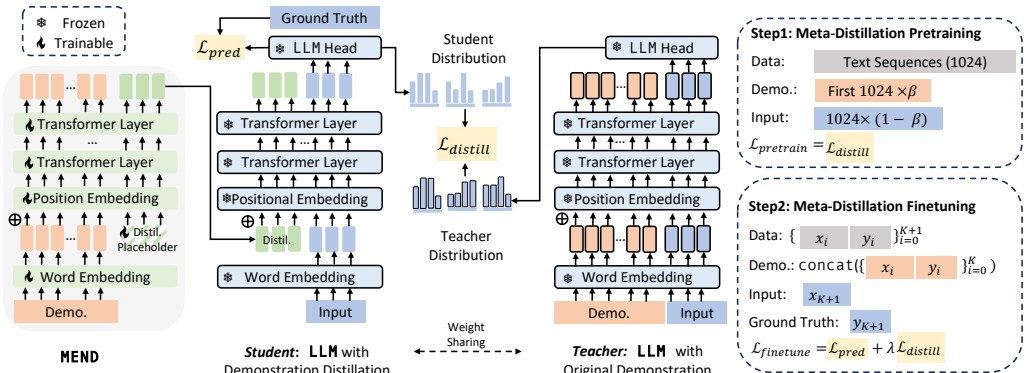

Figure 2: Overview of `MEND`. `MEND` takes as input demonstrations and distillation placeholder, outputs distillation vectors. To capture the meta-knowledge of demonstration distillation, `MEND` is trained in two stages: meta-distillation pretraining and fientuning.

probability for each label $c \in \mathcal{C}$ and return the maximum conditional probability as:

$$\operatorname{argmax}_{c \in \mathcal{C}} P_{\texttt{LLM}}(c|\texttt{concat}(\mathbf{E}_D, \mathbf{E}_x)), \tag{1}$$

where $\mathcal{C}$ is the unique set of $\{y_i\}_{i=1}^K$ in classification tasks or answer options in question answering tasks, and $\mathbf{E}_{(\cdot)}$ is `LLM`'s word embedding.

To improve the efficiency of `ICL`, many related works (Lester et al., 2021; Phang et al., 2023; Ivison et al., 2022; Wang et al., 2023; Mu et al., 2023) aim to reduce the demonstrations length for `LLM` from $|D|$ into $l$ such that $l << |D|$. They synthesize a high-fidelity demonstration summary $\mathbf{S}_D \in \mathbb{R}^{l \times d}$, where $d$ is the hidden size of word embedding, to replace $D$:

$$\operatorname{argmax}_{c \in \mathcal{C}} P_{\texttt{LLM}}(c|\texttt{concat}(\mathbf{S}_D, \mathbf{E}_x)). \tag{2}$$

Prompt tuning approaches (Lester et al., 2021; Wang et al., 2023) consider $\mathbf{S}_D$ as learnable parameters. However, for other tasks' demonstrations like $D'$, it requires additional training time to get $\mathbf{S}_{D'}$. `Hypernetwork` approaches (Phang et al., 2023; Ivison et al., 2022; Mu et al., 2023) including our `MEND` address the challenge of retraining for novel, unseen tasks. They achieve this by employing a demonstration distillation model, denoted as $M$, which produce distillation vectors: $\mathbf{S}_D = M(\hat{\mathbf{E}}_D)$ and $\mathbf{S}_{D'} = M(\hat{\mathbf{E}}_{D'})$. These vectors correspond to any arbitrary demonstrations $D$ and $D'$. Here $\hat{\mathbf{E}}_{(\cdot)}$ represent the word embedding derived from the demonstration distillation model. Notably, previous `Hypernetwork` methods has the compatibility issues with `LLM`, resulting in distillation vectors of suboptimal quality.

## 3 METHODS

The whole framework of `MEND` is illustrated in Fig. 2. We insert $l$ special tokens to the vocabulary set of distillation language model `MEND`, which act as placeholders for the demonstration distillation. For any demonstrations $D$, these placeholders embedding $\hat{\mathbf{E}}_\phi$ are appended to the demonstrations embedding $\hat{\mathbf{E}}_D$, fostering a versatile distillation strategy suitable for diverse tasks. After multiple transformer layers inside `MEND`, we can distill the information from lengthy $D$ to compact distillation vectors $\mathbf{S}_D = \texttt{MEND}\left(\texttt{concat}(\hat{\mathbf{E}}_D, \hat{\mathbf{E}}_\phi)\right)_{[-l:]}$ abbreivated as $\mathbf{S}_D = \texttt{MEND}(\hat{\mathbf{E}}_D)$.

### 3.1 KNOWLEDGE DISTILLATION

The goal to knowledge distillation is to use a concise demonstration summary, $\mathbf{S}_D$, such that the downstream `LLM` behaves similar (e.g. output close word distributions) to its version conditioned on the full demonstrations $D$. To realize this, we treat the `LLM` with full demonstration $D$ as the "teacher" and the version with only the demonstration summary $\mathbf{S}_D$ as the "student". Subsequently, we employ KL divergence to assess the difference between the word probability distributions of these two models.

$$\mathcal{L}_{\texttt{distill}} = \text{KL}\left(P_{\texttt{LLM}}(x|\mathbf{E}_D) \,||\, P_{\texttt{LLM}}(x|\texttt{MEND}(\hat{\mathbf{E}}_D))\right), \tag{3}$$

We opted for KL divergence as our distillation objective to ensure the student model does not produce outputs that are too different from the teacher model.

### 3.2 OPTIMIZATION

Throughout our two-stage optimization process, LLM remains frozen, assisting in backpropagating the gradient from the loss to MEND.

**Meta-distillation Pretraining.** To help MEND capture the general knowledge of distillation, we pretrain it on a text pretraining data-C4 (Raffel et al., 2019). As illustrated in the right segment of Fig. 2, we extract sequences of 1024 tokens from the pretraining dataset. This sequence is divided into two parts: the first $1024 \times \beta$ tokens as demonstrations $D$ and the remainder, $1024 \times (1 - \beta)$, as input $x$, where $\beta$ is the hyperparameter to control the length of demonstrations. We then apply the knowledge distillation approach to pretrain MEND. In contrast with the conditional language modeling objective, where LLM predicts subsequent content based on compressed tokens (Phang et al., 2023; Ivison et al., 2022), our demonstration distillation is trained by minimizing $\mathcal{L}_{\texttt{distill}}$ and aims to ensure the distillation model more accurately captures the intrinsic attributes of MEND. Consequently, it can offer a more faithful demonstration distillation. As evidenced in Tab. 2 and Tab. 4, our demonstration distillation consistently outperforms the traditional conditional language modeling CLM approach.

**Meta-distillation Finetuning.** During this stage, we finetune MEND using ICL relevant tasks, equipping it with the ability to interpret a task's semantics from its demonstrations. This ensures that MEND can effectively generalize to unseen demonstrations in the future. In each iteration, we choose a meta-training task and extract $K + 1$ demonstrations from it. The first $K$ demonstrations are concatenated into $D$, while the remaining pair, $(x_{K+1}, y_{K+1})$ is reserved for test input and output purpose. Similar to the pretraining phase, the demonstrations $D$ are fed into the distillation model MEND, yielding the demonstration distillation $\mathbf{S}_D$. The primary purpose of $S_D$ is to instruct the LLM in producing $y$ and guarantee that LLM operates as though it was condition on the original demonstrations. The formulation of finetuning is as follows:

$$\mathcal{L}_{\text{pred}} = \log P_{\texttt{LLM}} \left( y | \texttt{concat}(\mathbf{S}_D, \mathbf{E}_x) \right),$$
$$\mathcal{L}_{\text{finetune}} = \mathcal{L}_{\text{pred}} + \lambda \mathcal{L}_{\text{distill}}. \tag{4}$$

where $\lambda$ is the hyper-parameter to control the importance of distillation in finetuning.

## 4 EXPERIMENTS

### 4.1 EXPERIMENT SETTING

**Benchmarks.** In the section, to validate our methodology, we employ the MetaICL dataset introduced by Min et al. (2022a), designed for in-context learning scenarios. MetaICL builds upon existing few-shot datasets, such as CrossFit (Ye et al., 2021) and UnifiedQA (Khashabi et al., 2020). Notably, the MetaICL dataset is divided into two distinct partitions: meta-train and meta-test, with no overlap between them. This setting expect the model first trained on meta-train then evaluated on meta-test dataset. Our experiments encompass seven distinct meta-train and meta-test partitions[3] as outlined in Tab. 1. In ICL, the context length is directly proportional to the number of demonstrations. For instance, in the Class→Class, with 16 demonstrations, each demonstration's average length is 56.21 tokens. Consequently, during inference, the average context length extends to 899.36 tokens (calculated as 16 × 56.21) which will bring additional computation compared with no demonstrations length with 56.21.

Following MetaICL setup (Radford et al., 2019), we utilize whitespace to delineate input and output. In most of our experiments, we have preset the number of demonstrations to $K = 16$. For evaluating model performance, accuracy metrics are employed for classification tasks, while Macro-F1 is utilized for non-classification tasks. In partitions that encompass both classification and non-classification tasks (such as LR), we compute the average of Macro-F1 and accuracy to assess overall performance.

**Base Models.** To illustrate the adaptability of our proposed MEND framework, we assess its performance using various backbone large language model architectures, including decoder-only models, such as GPT2 (Radford et al., 2019), and encoder-decoder models, like T5 (Raffel et al.,

---

[3]The tasks and their corresponding abbreviations can be found in Appendix A.

2019)[4]. We initially experimented with using different architectures for `MEND` and find that the when `MEND` and `LLM` are from the same model family works best. Thus, for `GPT-2`, we choose gpt2-small[5], while for `T5` we select t5-small-lm-adapt[6].

**Baseline Methods.** We compare the performance of `MEND` against four primary groups of baseline methodologies: *1)* `Zero-shot`: This approach utilizes the `LLM` for direct zero-shot inference. *2)* `Vanilla ICL`: Here, we employ `LLM` for in-context learning by conditioning on a concatenation of $K$ randomly selected demonstrations. *3)* `PromptTuning` (Lester et al., 2021): This strategy offers an efficient approach to adapt `LLM` to new tasks without requiring full retraining. *4)* `HyperTuning`: Phang et al. (2023) employs a language model to distill demonstrations into condensed vectors using a conditional language modeling

Table 1: Statistics of seven different task partitions. Each row indicates meta-training/test task partitions.

| Setting | meta-train | | | Setting | meta-test | |
| | # task | Avg. Len. | | | # task | Avg. Len. |
|---|---|---|---|---|---|---|
| Class | 43 | 44.54 | | Class | 20 | 56.21 |
| non-Class | 37 | 91.45 | | | | |
| QA | 37 | 91.58 | | QA | 22 | 57.84 |
| non-QA | 33 | 72.50 | | | | |
| non-NLI | 55 | 54.51 | | NLI | 8 | 61.61 |
| HR | 61 | 82.44 | | LR | 26 | 35.31 |
| non-Para | 59 | 55.97 | | Para | 4 | 54.06 |

objective. For fairness, `PromptTuning` and `HyperTuning`, use same prompt lengths and hypermodel sizes equivalent to those used in `MEND`. Further details regarding hyperparameter settings and analysis can be found in Fig. 4.

## 4.2 EXPERIMENT RESULTS

**Effectiveness.** This section outlines the results from our experiments, as detailed in Tab. 2. We make the following observations: *Firstly*, the `zero-shot` approach predominantly underperforms, indicating that the inductive biases introduced during meta-training (`PromptTuning`), meta-testing (`Vanilla ICL`), or both (`HyperTuning` and `MEND`) enhance in-context learning. *Secondly*, when compared with `PromptTuning`, both `HyperTuning` and `MEND` demonstrate marked improvements. This underscores the effectiveness and generalizability of using hypernetworks to distill the supervising signal from demonstrations to assist `LLM`. A potential reason for `PromptTuning`'s inferior performance is that it solely captures inductive bias through gradient descent during meta-training and cannot leverage bias from the meta-test's demonstrations at meta-test time. *Thirdly*, `Vanilla ICL` outperforms `HyperTuning`, while `MEND` consistently matches or even surpasses `Vanilla ICL`. This suggests that our approach, incorporating $\mathcal{L}_{\texttt{distill}}$ and $\mathcal{L}_{\texttt{pred}}$, is adept at capturing the meta-knowledge facilitating the distillation demonstration to aid `LLM`.

**Inference Efficiency.** Inference efficiency remains a fundamental aspect of our study. The core idea of our work is to distill extensive natural language demonstrations, denoted as $D$, into concise distillation vectors, denoted as $\mathbf{S}_D$, thereby reducing computational demands for `LLM`. To assess the efficiency of our model, we report the computational costs associated with different representation techniques in terms of processing time, memory consumption, and floating-point operations per second (FLOPS). Specifically, for each meta-test partition, we select a single task, evaluate it with a batch size of 1, and measure the aforementioned metrics. Considering that `HyperTuning` operates identically to `MEND` during inference, we have chosen `Vanilla ICL` and `PromptTuning` as our baseline methods. It is important to note that the inference efficiency of `MEND` encompasses both the process of obtaining the distilled vectors and the subsequent inference by the `LLM` using these vectors in conjunction with the test input. Compared with `PromptTuning`, `MEND` bring additional computational cost at compressing demonstrations into compact vectors. As illustrated in Fig. 3, `MEND` achieves up to 3.5 times greater computational efficiency compared to `Vanilla ICL` and requires less peak GPU memory. Remarkably, while `MEND` demonstrates efficiency on par with `PromptTuning`, it also presents a notable performance enhancement, as evidenced in Tab. 2. These observations indicate our proposed method `MEND` can improve the `LLM`'s efficiency without sacrificing `LLM`'s effectiveness in in-context learning.

---

[4]In § C, we have test our proposed method on flat-t5-xl and opt-6.7b.
[5]https://huggingface.co/gpt2
[6]https://huggingface.co/google/t5-small-lm-adapt

Table 2: Performance on the `MetaICL` Dataset: This table shows the average and stand deviation scores from running our evaluation with five distinct random seeds. To enhance readability, we present the meta-train and meta-test pairs in the format "meta-train → meta-test". The best-performing models are highlighted in bold, while the second-best are underlined. The standard deviation values reflect the variability due to different demonstrations retrieved. Note that the "PromptTuning" and "zero-shot" approaches do not require demonstration retrieval, hence their standard deviation is zero.

| Methods | Class → Class | non-Class → Class | non-NLI → NLI | non-QA→QA | QA → QA | HR → LR | non-Para→Para | AVG |
|---|---|---|---|---|---|---|---|---|
| gpt2-large | | | | | | | | |
| zero-shot | 34.36 | 34.36 | 25.50 | $\underline{44.58}$ | 44.58 | 34.77 | 34.12 | 36.04 |
| PromptTuning | 37.65 | 38.78 | 31.34 | 38.71 | 45.77 | 40.68 | 34.23 | 38.17 |
| Vanilla ICL | $41.30_{\pm2.15}$ | $41.30_{\pm2.15}$ | $39.13_{\pm2.30}$ | $\mathbf{45.81}_{\pm1.34}$ | $45.81_{\pm1.34}$ | $\underline{41.26}_{\pm2.26}$ | $38.93_{\pm1.15}$ | $\underline{41.93}$ |
| HyperTuning | $40.42_{\pm1.64}$ | $42.54_{\pm1.79}$ | $36.49_{\pm2.01}$ | $41.11_{\pm0.82}$ | $\underline{46.20}_{\pm0.50}$ | $\mathbf{41.63}_{\pm1.72}$ | $\underline{39.63}_{\pm0.66}$ | 41.15 |
| MEND | $\mathbf{43.35}_{\pm2.17}$ | $\mathbf{43.38}_{\pm1.62}$ | $\mathbf{39.96}_{\pm1.99}$ | $44.29_{\pm0.86}$ | $\mathbf{46.92}_{\pm0.49}$ | $40.92_{\pm1.80}$ | $\mathbf{42.54}_{\pm0.44}$ | $\mathbf{43.05}$ |
| gpt2-xl | | | | | | | | |
| zero-shot | 32.08 | 32.08 | 25.54 | $\underline{46.09}$ | 46.09 | 33.95 | 33.61 | 35.63 |
| PromptTuning | 37.65 | 38.78 | 36.27 | 41.45 | 46.95 | 40.83 | 35.52 | 39.64 |
| Vanilla ICL | $\underline{40.63}_{\pm2.53}$ | $40.63_{\pm2.53}$ | $\mathbf{37.35}_{\pm1.83}$ | $\mathbf{48.32}_{\pm0.88}$ | $\mathbf{48.32}_{\pm0.88}$ | $\underline{42.27}_{\pm2.08}$ | $\underline{37.53}_{\pm1.04}$ | $\underline{42.15}$ |
| HyperTuning | $40.26_{\pm1.33}$ | $\mathbf{43.74}_{\pm1.51}$ | $34.61_{\pm1.23}$ | $40.71_{\pm1.14}$ | $47.41_{\pm0.46}$ | $41.83_{\pm1.34}$ | $35.72_{\pm0.43}$ | 40.61 |
| MEND | $\mathbf{42.79}_{\pm2.22}$ | $\underline{43.37}_{\pm1.50}$ | $\underline{37.00}_{\pm1.99}$ | $45.95_{\pm0.66}$ | $\underline{48.07}_{\pm0.40}$ | $\mathbf{42.16}_{\pm1.81}$ | $\mathbf{42.53}_{\pm1.20}$ | $\mathbf{43.12}$ |
| t5-lm-large | | | | | | | | |
| zero-shot | 36.75 | 36.75 | 25.72 | 39.05 | 39.05 | 32.09 | 34.28 | 34.81 |
| PromptTuning | 32.56 | 32.37 | 25.80 | $\underline{39.48}$ | 39.44 | 32.43 | 36.44 | 34.07 |
| Vanilla ICL | $\underline{38.40}_{\pm2.87}$ | $\underline{38.40}_{\pm2.87}$ | $\underline{36.68}_{\pm2.37}$ | $39.26_{\pm1.23}$ | $39.26_{\pm1.23}$ | $\underline{38.77}_{\pm2.13}$ | $36.31_{\pm0.51}$ | $\underline{38.15}$ |
| HyperTuning | $31.17_{\pm2.46}$ | $29.06_{\pm1.96}$ | $33.56_{\pm1.76}$ | $39.03_{\pm1.09}$ | $\underline{41.17}_{\pm0.86}$ | $34.28_{\pm1.27}$ | $\underline{37.39}_{\pm2.67}$ | 35.09 |
| MEND | $\mathbf{41.75}_{\pm1.82}$ | $\mathbf{38.93}_{\pm1.43}$ | $\mathbf{37.15}_{\pm2.00}$ | $\mathbf{41.76}_{\pm0.60}$ | $\mathbf{42.91}_{\pm0.55}$ | $\mathbf{39.07}_{\pm2.16}$ | $\mathbf{36.99}_{\pm0.55}$ | $\mathbf{39.79}$ |

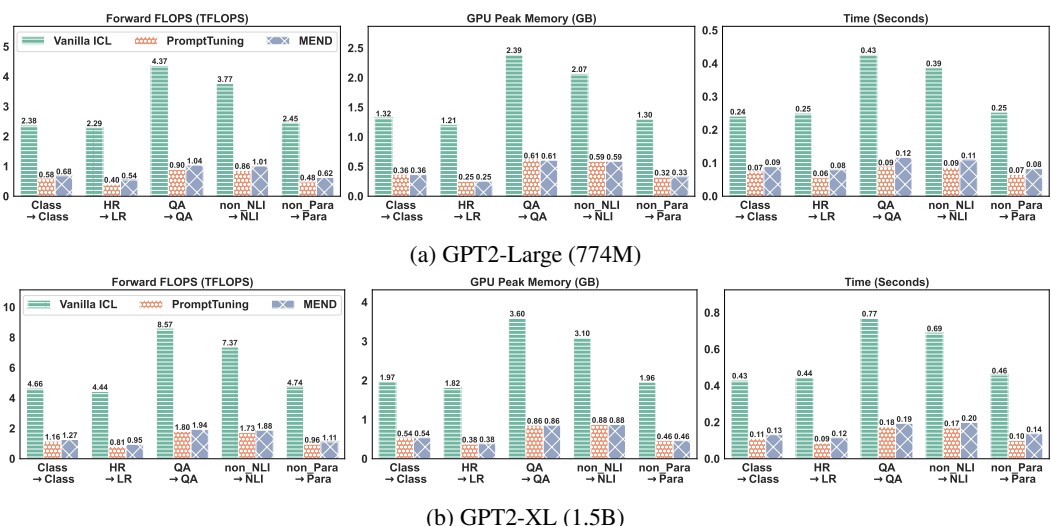

(a) GPT2-Large (774M)

(b) GPT2-XL (1.5B)

Figure 3: **Efficient Analysis of In-Context Learning at Inference Time.** GPT2-large (774M) and GPT2-XL(1.5B) are evaluated on the same task with batch size 1. The context length for both `PromptTuning` and `MEND` is 100, while for `Vanilla ICL` varies on the partitions. (Class→Class is 469, HR→LR is 652, QA→QA is 639, non_NLI→NLI is 848, and non_Para→Para is 818).

## 5 ANALYSIS

In this section, we conduct a comprehensive examination of our distillation approach across various scenarios to gain deeper insights into its behavior and potential limitations. To mitigate computational resource demands, we primarily employ the gpt2-large model as `LLM` on Class→Class setting unless mentioned otherwise.

## 5.1 VARYING DEMONSTRATION DISTILLATION RATIO

A crucial aspect of our experimental analysis was to comprehend how varying the demonstration distillation ratio impacts the distillation of demonstrations and, consequently, the effectiveness of LLM's in-context learning. The demonstration distillation ratio is defined as the ratio of the number of demonstrations to the length of distillation vectors. Specifically, we vary the distillation ratio from two perspectives: the richness of input (the number of demonstration examples) and the compactness of the output (the length of demonstration distillation).

**Varying Number of Demonstrations.**    We assess the effectiveness of our method while altering the value of $K$ (the number of demonstration) while keeping the length of the distillation vector $l$ constant. As depicted in Fig. 4a, our `MEND` approach consistently outperforms the `Vanilla ICL` and `HyperTuning` methods for various values of K (1, 2, 4, 8, and 16). Furthermore, `MEND` demonstrates consistent performance improvement as K increases, whereas `Vanilla ICL` reaches its peak performance at $K = 4$. This improvement suggests that `MEND` is excels at extracting supervision information for in-context learning from the selected demonstration examples.

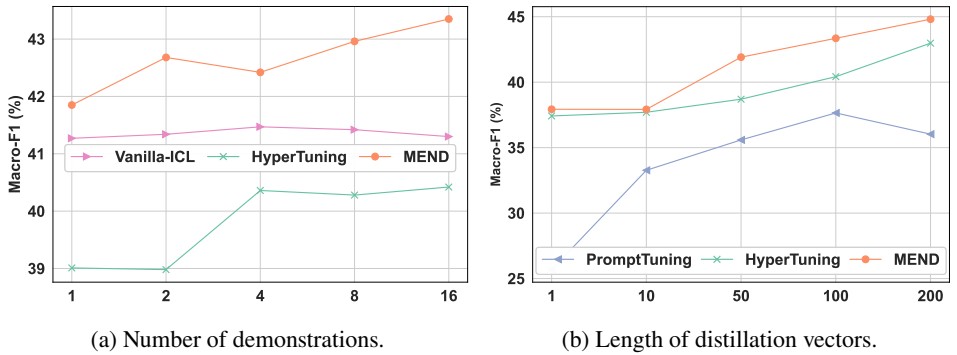

(a) Number of demonstrations.                    (b) Length of distillation vectors.

Figure 4: Performance with different demonstration distillation ratio. The distillation ratio is the ratio of the number of demonstration examples to the length of the distillation.

**Varying demonstration distillation Length.**    We manipulate the length of demonstration distillation $l = 1, 10, 50, 100$ and $200$ while keeping $K = 16$. It is worth noting that we retrain `MEND` with two stages as shown in § 3.2 for different $l$ values. The results in Fig. 4b yield the following observations: *Firstly*, as the demonstration distillation length increases, the performance of all methods generally improves, except for $l = 200$ in the case of `PromptTuning`. This suggests that there may be information loss in demonstration distillation, and increasing the length of the demonstration may help mitigate this issue. However, there exists a trade-off between efficiency and effectiveness, as extending the length of the distillation vectors results in a quadratic time complexity increase. *Secondly*, we observe that our proposed method achieves the best performance among the baseline methods, including `HyperTuning`. This underscores the significance of our optimization design in providing enhanced inductive bias for in-context learning.

## 5.2 PERTURBATION TO DEMONSTRATIONS

Given the significant influence of provided demonstrations on the performance of in-context learning (Min et al., 2022b), we aim to investigate whether our proposed approach, `MEND`, can effectively distill and propagate modifications made to demonstrations to the distilled vectors. To address this, we empirically perturb the demonstrations from both positive and negative perspectives.

**Positive Perturbation.**    In light of previous research Liu et al. (2021) emphasizing the value of semantically similar demonstrations and their positive impact on in-context learning, we aim to ascertain whether `MEND`'s advantages are complemented by or enhanced through the use of improved retrieved demonstrations. We transit from a random sampling approach to a more nuanced semantic-based $k$-NN retrieval method. As indicated in Tab. 3, semantic-based retrieval methods, including *dense* and *bm25*, exhibit superior performance compared to random selection under the *No Perturbation* condition. Remarkably, `MEND` not only matches or even surpass the performance of these advanced retrieval methods and does so with a reduced context size.

Table 3: Performances when applying perturbations on demonstrations.

| Methods | No Perturbation | Positive Perturbation | | Negative Perturbation | | | |
|---|---|---|---|---|---|---|---|
| | | bm25-$k$NN | dense-$k$NN | No Label | No Input | Random Label | Wrong Label |
| Vanilla ICL | 41.30 | 45.38 | 48.33 | 30.57 | 42.29 | 37.25 | 28.13 |
| HyperTuning | 40.42 | 43.95 | 45.13 | 31.78 | 38.20 | 38.72 | 29.31 |
| MEND | **43.35** | **46.82** | **48.81** | **32.57** | **44.29** | **39.25** | **30.42** |

**Negative Perturbation.** We evaluate the impact of various negative perturbations, including the following scenarios: *1) No Label*: This perturbation involves removing the labels while retaining the inputs. *2) No Input*: The inputs are removed while keeping the labels intact. *3) Random Label*: This perturbation randomly selects one of the valid options as the output. *4) Wrong Label*: In this case, one of the incorrect options is randomly selected. The results are presented in Tab. 3. As anticipated, a consistent trend emerges, with *No Perturbation* outperforming both *Random Label* and *Wrong Label* for both the Vanilla ICL and our proposed MEND. Moreover, it is noteworthy that performance improves in most cases when the *No Input* perturbation is applied. This not only underscores the significance of labels in the context of in-context learning but also illustrates MEND's ability to effectively distill label information into the distilled vectors.

## 5.3 ATTENTION WEIGHT VISUALIZATION

To gain a deeper understanding of how demonstration distillation impacts LLM, we employ visualization techniques to explore the attention weights of LLM's induction heads, as introduced by Olsson et al. (2022). Induction heads are attention heads known for their prefix matching and copying properties, which play a crucial role in the context of in-context learning. They empirically increase the likelihood of $[B]$ given $[A][B]\cdots[A]$ when repeated sequence of tokens. Our objective is to understand whether our demonstration distillation can store the input-output pattern that will activate these induction heads in a manner similar to the original demonstration tokens.

We visualize the attention weights of the four induction heads[7] for both Vanilla ICL and MEND, as illustrated in Fig. 5. A review of Fig. 5 reveals that the final prediction establishes a constructive association with the demonstration distillations. Given that the length of demonstration tokens (average=914) and compressed prompt tokens (100) significantly exceed the length of test input, we employ max pooling to map the attention weights of the demonstrations into 20 tokens (Area enclosed by red rectangle). This in-depth analysis further substantiates that the distillation derived from MEND offers valuable context supervision signals for LLM.

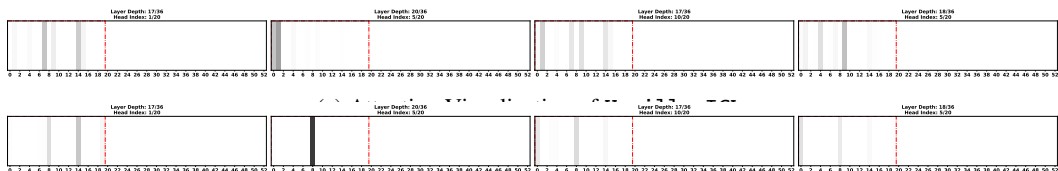

(b) Attention Visualization of MEND

Figure 5: Attention visualization. The left red surrounded x-axis denotes either the demonstrations (Vanilla ICL) or the distilled vectors (MEND) and the other part of x-axis are the tokens from the test input. The y-axis corresponds to the first token of the output word.

## 5.4 ABLATION STUDY ON DEMONSTRATION DISTILLATION

To assess the significance of the $\mathcal{L}_{distill}$, we conducted an experiment that excluding this term during both the pretraining and finetuning stages on several representative task paritions.

**Pretraining.** During the pretraining phase, we compare using no-pretraining, conditional language modeling (CLM) (Phang et al., 2023), and CLM+$\mathcal{L}_{distill}$[8]. We find that (1) pretraining is crucial as it substantially enhances performance compared to methods with no-pretraining, except for the no-pretraining baseline; (2) our pretraining approach outperforms the alternatives. We hypothesize that this superiority is attributed to our pretraining scheme better align the MEND and LLM.

---

[7]The details of identifying induction heads can be found in § C.

[8]More analysis about CLM+$\mathcal{L}_{distill}$ can be found in Appendix B

**Finetuning.** In this phase, we retained the same pretraining objective function but omitted various finetuning components. Examining the lower section of Tab. 4, we observe that the removal of each component leads to a decrease in performance. This observation underscores the positive contributions of each component within our proposed method to the overall performance.

Table 4: Ablation study of knowledge distillation.

| Methods | non-Class → Class | non-NLI → NLI | non-QA→QA | QA → QA | Avg. |
|---|---|---|---|---|---|
| Vanilla ICL | 41.30 | 39.13 | **45.81** | 45.81 | 43.01 |
| MEND | **43.38** | **39.96** | 44.29 | **46.92** | **43.65** |
| Ablation Study on Pretraining | | | | | |
| No-Pretraining | 38.25 | 34.33 | 42.18 | 45.65 | 40.10 |
| CLM | 42.00 | 39.09 | 44.13 | 46.47 | 42.92 |
| CLM + $\mathcal{L}_{distill}$ | 41.38 | 38.79 | 43.69 | 45.10 | 42.24 |
| Ablation Study on Finetuning | | | | | |
| MEND w/o $\mathcal{L}_{pred}$ | 37.64 | 33.90 | 43.97 | 44.54 | 40.41 |
| MEND w/o $\mathcal{L}_{distill}$ | 39.26 | 37.22 | 40.29 | 45.78 | 40.64 |

In this experiment, we also observed that both the pretraining and finetuning ablations of MEND significantly underperform compared to Vanilla ICL. This finding underscores the critical role of the two-stage design, encompassing both pretraining and finetuning, in our model's effectiveness. Moreover, it highlights the essential contribution of knowledge distillation in replicating the teacher model's behaviors and harnessing meta-training knowledge. These results collectively illustrate the synergistic impact of these components in enhancing MEND's performance.

## 6 RELATED WORK

**Hypernetwork** The concept of a Hypernetwork, as introduced by Ha et al. (2016), refers to an auxiliary network designed to generate parameters for a primary network. In a similar view, MEND can be perceived as a Hypernetwork, producing distilled vectors (parameters) to tailor LLM for new tasks. Notable efforts like HyperTuning (Phang et al., 2023), HINT (Ivison et al., 2022), Hyper(Ye & Ren, 2021) have employed a language model-based distillation model to condense demonstrations into distilled vectors. While these methods can adapt to unseen demonstrations, they often degrade with ICL performance. On the other hand, Gist (Mu et al., 2023) enhances the LLM with instruction distillation and instruction following. However, given that the distillation model is synonymous with the LLM, the distillation procedure induces computational overhead, especially when compared with our approach that deploys a smaller language model for distillation. A distinctive advantage of MEND over existing Hypernetwork-based demonstration distillations is its simultaneous realization of efficiency and effectiveness as shown in Tab. 2 and Fig. 3.

**Knowledge Distillation** Knowledge distillation, as introduced by Hinton et al. (2015), seeks to transfer insights from a high-capacity model to a model with lower capacity. This methodology is key in ensuring both efficiency and effectiveness for MEND, setting MEND apart from other HyperNetwork techniques. Askell et al. (2021); Snell et al. (2022) exploit the knowledge distillation to finetune LLM with the ability to function as the language model with a prepended prompt when did not provide any prompt. Nonetheless, given the diverse nature of demonstrations, as illustrated in Tab. 3, these methods fail to include superior demonstrations for better ICL performance. Furthermore, as MEND functions as a complementary module for LLM, it doesn't hamper LLM's inherent capabilities Furthermore, as MEND functions as a complementary module for LLM, it doesn't hamper LLM's inherent capabilities.

## 7 CONCLUSION

We introduced MEND to not only tackle the inherent efficiency challenges in in-context learning with large language models but also to address the effectiveness limitations of existing demonstration distillation methodologies. Our innovative approach distilled in-context demonstrations into vectors, tailored for downstream large language models. Rigorous evaluations of MEND across seven distinct few-shot task partitions and two major large language model families have underscored its prowess. Notably, MEND consistently matches or even surpasses the performance of traditional in-context learning, all while demanding fewer FLOPs. This breakthrough paves the way for more efficient and scalable applications of large language models in real-world scenarios. In the future, we aim to distill an even broader spectrum of demonstrations, some potentially surpassing the context window limits of both the demonstration distillation model and LLM.

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

## A  DATA, TRAINING, EVALUATION, AND COMPUTE DETAILS

Code and data are available in the supplementary material and will be made public upon paper acceptance via GitHub.

**Data.**  For pretraining stage, we utilize the C4 validation dataset (Raffel et al., 2019) as our training data. We truncate each passage into 1024 tokens. For meta-distillation stage, we limit the context length into 900. Within the demonstrations, any example $\mathbf{x}_i$ exceeding 256 tokens is truncated from the end. However, we do not truncate the label $\mathbf{y}_i$. If the context length surpasses 900 tokens while $i < K$, the subsequent demonstrations $\{(\mathbf{x}_{i+1}, \mathbf{y}_{i+1})\}^K$ are omiited.

The tasks and their corresponding abbreviations are as follows: "Class" for classification, "QA" for question answering, "NLI" for natural language inference, "HR" for high resource, "LR" for low resource, and "Para" for paraphrase.

**Training.**  The complete set of stable hyperparameters for training runs can be found in Tab. 5. These parameters are adapted from `MetaICL` (Min et al., 2022a). Additional hyperparameters that needed exploration and their corresponding search spaces are also detailed in Tab. 5.

For *pretraining*, we leverage the Class→Class meta-test validation dataset for early stopping. It should be noticed that while determining pretraining hyperparameters, we focused our search solely on gpt2-large and subsequently adapted the findings to other downstream `MEND`.

As for `finetuning`, we use specific meta-test validation data for early stopping. When it comes to the meta-distillation finetuning hyperparameters, we conduct the search for each task split and `MEND` independently.

The hyperparameter analysis of $\beta$ and $\lambda$ can be found in Fig. 7 and Fig. 6a.

Table 5: Hyperparameters for `MEND`.

|  | Pretraining | | | Finetuning | | |
|---|---|---|---|---|---|---|
|  | gpt2-large | gpt2-xl | t5-large-lm | gpt2-large | gpt2-xl | t5-large-lm |
| Stable Hyperparameters | | | | | | |
| num steps | 30,000 | 30,000 | 5,000 | 30,000 | 30,000 | 30,000 |
| batch size | 1 | 1 | 8 | 1 | 1 | 1 |
| learning rate | 5e-5 | 5e-5 | 5e-5 | 5e-5 | 5e-5 | 5e-5 |
| precision | fp16 | fp16 | fp32 | fp16 | fp16 | fp32 |
| optimizer | adamW | adamW | adamW | adamW | adamW | adamW |
| LLM$_\theta$ in 8bit | True | True | False | True | True | False |
| early stop patience | 5 | 5 | 5 | 5 | 5 | 5 |
| Searchable Hyperparameters | | | | | | |
| $\beta$ | | $[0.1, 0.5, 0.8, 0.9]$ | | N/A | N/A | N/A |
| $\lambda$ | N/A | N/A | N/A | | $[0.01, 0.1, 1, 10]$ | |

**Compute.**  We implemented our proposed methodology using PyTorch v1.13.1 (Paszke et al., 2019), complemented by the HuggingFace Transformers library v4.24.0 (Wolf et al., 2019) and Accelerate v0.20.0 (Gugger et al., 2022). All experiments were conducted on eight A10 NVIDIA GPUs, each equipped with 24GB of memory.

## B  HYPERPARAMETER ANALYSIS

**Pretraining relevant Hyperparameters.**  During the pretraining stage, there are two important factors greatly influence the distillation models performance for the following Meta-Distillation fineuning: $\beta$ and $\gamma$. $\beta$ controls the length of demonstrations for distillation during pretraining and $\gamma$ controls the importance of knowledge distillation during pretraining. In Tab. 4, we show the experiment results of `CLM+1` $\times \mathcal{L}_{\text{distill}}$). To comprehensively understand the superiority of sole $\mathcal{L}_{\text{distill}}$, we consider an the hyperparameter analysis on the combination of `CLM+1` $\times \mathcal{L}_{\text{distill}}$, which can be formulated as $\mathcal{L} = \mathcal{L}_{\text{CLM}} + \gamma \mathcal{L}_{\text{distill}}$. To save computational resource, different from Tab. 4 we directly report the experiment result after pretraining without further Meta-distillation comprehension.

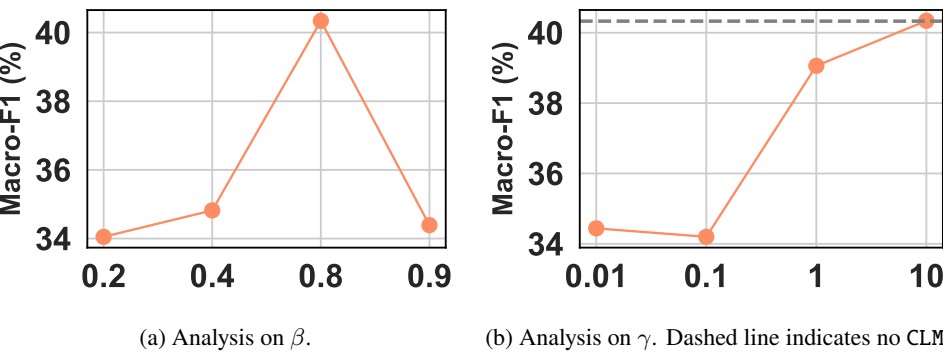

(a) Analysis on $\beta$.          (b) Analysis on $\gamma$. Dashed line indicates no `CLM`.

Figure 6: Analysis on pretraining relevant hyperparameters.

As the result shown in Fig. 6, we have the following observations: *1)* `MEND` achieves the best performance when $\beta = 0.8$. This indicates that during pretraining, proper design the ratio of demonstrations to inputs will achieve better performance than small or large ratios; *2)* `MEND` achieves better performance when increasing the $\gamma$. This indicates the importance of $\mathcal{L}_{distill}$ (knowledge distillation) in minimize the knowledge gap between the distillation model and downstream language model.

**Meta-Distillation relevant Hyperparameters.**
To understand the importance of knowledge distillation in Meta-distillation finetuning stage, we vary $\lambda$ in Eq. 4. As the result shown in Fig. 7, we can observe that `MEND` achieve beter performance when $\lambda >= 1$, this also indicates the importance of knowledge distillation.

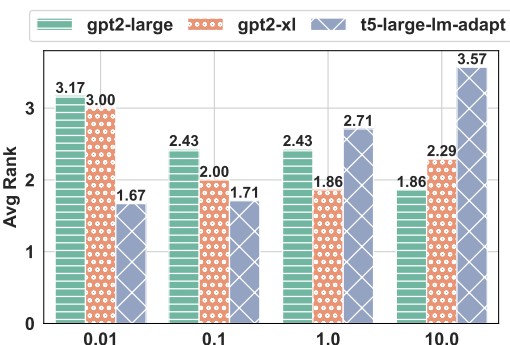

Figure 7: Hyperparameter analysis on $\lambda$.

## C   ADDITIONAL ANALYSIS

**Identify Induction Head.**   In § 5.3, we visualize the attention weights of induction heads. Here, we introduce how we identify these induction heads. Following (Olsson et al., 2022; Nanda & Bloom, 2022), we firstly create 10 randomly sequences with length 500 then expand them by concatenating with itself for time. Thus we have 10 sequences with length 1000 and for each sequence, the first 500 tokens is exact same as the rest 500 tokens. Then, inside each self-attention layer, we take the diagonal of attention paid from each destination position (position index $> 500$) to source positions $500 - 1$ back and get the attention average of each head over these tokens. The average attention score are shown in Fig. 8 We choose the 4 attention head with largest average attention score as the our interested inductive head.

**Additional Large Language Model.**   To assess the efficacy and generalizability of `MEND`, we conducted evaluations on larger models, specifically `opt-6.7b` Zhang et al. (2022) and `flan-t5-xl` Chung et al. (2022). For demonstration distillation, we strategically selected smaller counterparts as backbone models: `opt-125m` for `opt-6.7b` and `flan-t5-base` for `flan-t5-xl`. We maintained consistent formatting and training methodologies across these evaluations, using whitespace to separate inputs and outputs within and across demonstrations, as done with gpt2-large. The results, as detailed in Tab. 6, show that `MEND` consistently outperforms other baseline methods. This demonstrates its ability to effectively capture and utilize meta-knowledge, enhancing the efficiency of demonstration distillation for aiding large language models (`LLM`).

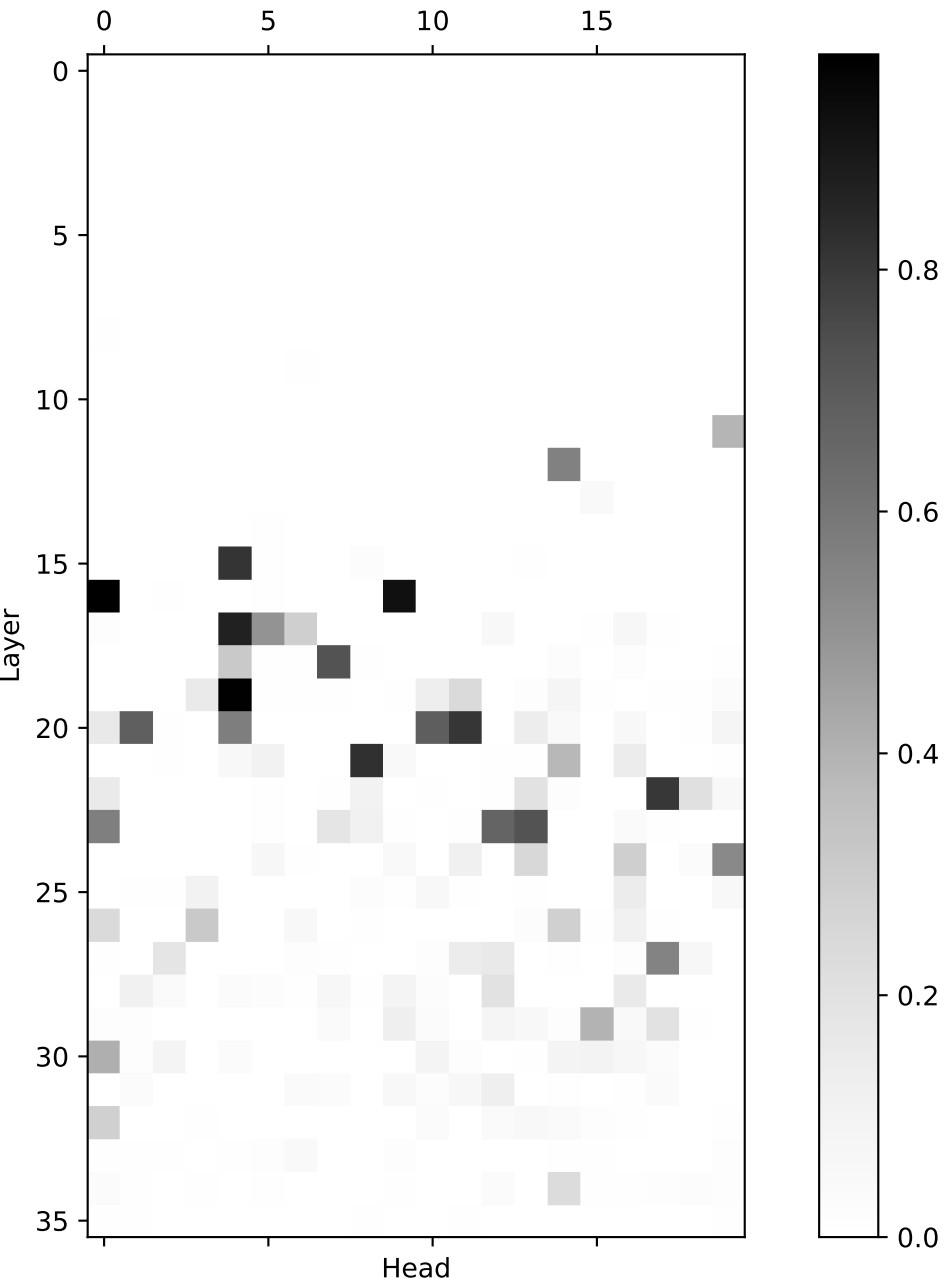

Figure 8: Average attention weight visualization of attention head from gp2-large .

**Robustness towards Template Variations**
While the primary objective of our study is to distill demonstrations into compact vectors, the exploration of optimal prompt templates is beyond the scope of this paper. In our experiments, we consistently used whitespace to separate inputs and outputs within and between demonstrations across all models. To assess the robustness of our models against template variations, we conducted an additional evaluation. We transferred the model trained with a whitespace separator to a new template using newline characters $(nn)$ for separating inputs and outputs, and three newlines for differentiating between demonstrations on the `gpt2-large` LLM. The results, presented in Tab. 7, indicate that `MEND` exhibits minimal sensitivity to these format changes. The performance difference was negligible, with less than a 0.3% variance between using spaces and newlines.

Table 6: Experiment on advanced large language models.

| Methods | Class $\rightarrow$ Class |
| --- | --- |
| flan-t5-xl | |
| PromptTuning | 33.24 |
| Vanilla ICL | $40.63_{\pm 2.21}$ |
| HyperTuning | $39.70_{\pm 1.38}$ |
| MEND | $\mathbf{40.77}_{\pm 1.20}$ |
| opt-6.7b | |
| PromptTuning | 38.81 |
| Vanilla ICL | 42.38 |
| HyperTuning | $32.67_{\pm 2.17}$ |
| MEND | $\mathbf{44.27}_{\pm 1.12}$ |

## D    LIMITATIONS

**Large Downstream language Models.** Due to computational constraints, our experiments use models that are <2B. Whether these demonstration language distillation techniques generalize o the largest models (10B+) is unknown. However, given that our method can generalize to different model structures and computation efficiency without hurting the downstream language model's performance, we believe we are shedding insights for future work.

Table 7: Robustness of template variations. All the method is evaluated on Class $\rightarrow$ Class setting. The Diff. is the difference between newline result minus whitespace result.

| Methods | whitespace | newline | Diff. |
| --- | --- | --- | --- |
| Vanilla ICL | $41.30_{\pm 2.15}$ | $38.90_{\pm 2.21}$ | $-2.40$ |
| HyperTuning | $40.42_{\pm 1.64}$ | $40.08_{\pm 2.54}$ | $-0.34$ |
| MEND | $43.35_{\pm 2.17}$ | $43.50_{\pm 2.12}$ | $+0.15$ |

**Language Model dependent.** Due to our design of distillation, the `MEND` may face the adaptation problem across different `MEND`s. This means we need to train a new distillation model for any new `LLM`. In addition, because of our optimization design, we need the gradients that back propagate on the top of `MEND`s. This will bring computation overhead when we try large `LLM` with larger demonstration encoders.

**Limited Context Window.** Both `MEND` and `LLM` have a limited context window. Thus, when demonstrations exceeds the length context, we inevitably need to truncate the demonstration. This will not only lose the information from the discarded tokens and cannot distill large amount of demonstration(e.g. $K > 1000$ (Hao et al., 2022)). Concurrent work utilizes recurrent memory transformer (Bulatov et al., 2022) to compress long text documents beyond the constraint of context window size into soft prompts. We consider handling extra-long demonstration as our future work.

