# OpenReview forum: "MEND: Meta Demonstration Distillation for Efficient and Effective In-Context Learning"
_ICLR.cc/2024/Conference — ICLR 2024 poster_

### Official Review · Reviewer_RPCQ · 2023-10-29

**Soundness:** 3 good
**Presentation:** 2 fair
**Contribution:** 2 fair
**Rating:** 6
**Confidence:** 4

**Summary:**

This paper proposes a demonstration compression framework to save retraining LLMs for unseen tasks under in-context scenarios. To be specific, a two-stage training process including knowledge distillation from pre-trained LLMs and fine-tuning on specific tasks endows the framework with both efficiency and effectiveness. Empirical results of decoder-only and encoder-decoder architectures validate the proposed method.

**Strengths:**

- The authors proposed an efficient demonstration distillation for in-context learning.
- The paper is well-organized and easy to follow.

**Weaknesses:**

- More insightful explanations about why the proposed method does not compromise the in-context learning ability of LLMs could make the paper stronger.
- self-contained notations may help the readers to understand the results better.
- Figure 5 can be displayed using more contrasting colors

**Questions:**

- The distillation loss occurs both in the pretraining and fine-tuning stages where a lambda controller balances the influence of distillation. What’s the influence of the lambda? A clear explanation about connecting the training mechanism(two-stage process) and each loss term to the “require task-specific retraining or compromise in-context learning” would help the reviewers better understand the advantage of the method.

- In Figure 3, is the size of the distillation vectors of MEND the same size as PromptTuning?
What’s the model size of MEND and FLOPs it introduced?


Several questions for Table 4:
- In the ablation of pre-training,  even if the row is labeled as “No pretraining”, the fine-tuning term still contains the distillation loss. So what’s the lamda for this row?
- What does L_hidn mean in Table 4? Is it L_distill ?
  If it is L_distll, then the second to the last row is the result of pretraining loss for both stages. The performance looks far from comparable to CLM. Can the authors clearly explain this?

---

> ### Author Response · Authors · 2023-11-23
> **Response to Reviewer RPCQ (Part 1)**
>
> **Q1. More insightful explanations about why the proposed method does not compromise the in-context learning ability of LLMs could make the paper stronger.**
>
> **A1.** Thank you for the opportunity to elaborate on why our proposed method, MEND, does not compromise the in-context learning ability of LLMs. The key to this lies in our novel two-stage training procedure.
> - Firstly, our pretraining stage is designed to align the condensed distillation vectors with the word embeddings of the LLM. As evidenced in Table 4, variations without this specialized pretraining show inferior performance compared to vanilla ICL, highlighting the pretraining's role in ensuring the effectiveness of the distilled vectors.
> - Secondly, the fine-tuning stage leverages objective functions that are crucial for capturing the essence of in-context learning. Removing any of these objectives results in a noticeable performance drop, underscoring their importance. This stage essentially harnesses meta-knowledge from both the data and the teacher model, ensuring that MEND can adeptly adapt to new tasks during testing.
> - Furthermore, our approach is supported by findings in relevant literature [1-3], which demonstrate the effectiveness of similar methodologies in enhancing LLMs' learning capabilities. This dual-stage training process is integral to MEND’s ability to maintain, and often enhance, the in-context learning performance of large language models without significant computational overhead."
>
> [1] Min, Sewon, Mike Lewis, Luke Zettlemoyer, and Hannaneh Hajishirzi. "Metaicl: Learning to learn in context." arXiv preprint arXiv:2110.15943 (2021).
>
> [2]Ye, Qinyuan, Iz Beltagy, Matthew E. Peters, Xiang Ren, and Hannaneh Hajishirzi. "FiD-ICL: A Fusion-in-Decoder Approach for Efficient In-Context Learning." In Proceedings of the 61st Annual Meeting of the Association for Computational Linguistics (Volume 1: Long Papers), pp. 8158-8185. 2023.
>
> [3] Phang, Jason, Yi Mao, Pengcheng He, and Weizhu Chen. "Hypertuning: Toward adapting large language models without back-propagation." In International Conference on Machine Learning, pp. 27854-27875. PMLR, 2023.
>
> **Q2. More contrast color for Figure 5.**
> **A2.** Thank you for your valuable suggestion regarding the visualization in Figure 5.
> We originally chose a grayscale color scheme to represent the attention map due to the wide range of values, from $10^{-1} to 10^{-8}$ and the focused distribution of attention weights on specific tokens. This resulted in many tokens having very small attention weights, which are represented in lighter colors close to white.
> To enhance the readability of the figure without compromising its informational content, we are considering not only the addition of x-ticks to better differentiate between tokens from demonstrations. This adjustment will make it easier to discern the differences in attention weights across tokens.
> Furthermore, it is crucial to note the significance of Figure 5 in our paper.
> It illustrates that even though the demonstrations are condensed, our model, MEND, can still effectively extract and utilize the necessary information from the distilled vectors for accurate label prediction. This highlights the efficacy of MEND in maintaining key information despite significant data compression, a central aspect of our work.
>
>
>
> **Q3.1. Influence of lambda in controlling the importance of distillation loss in meta-training?**
>
> **A3.1.** hank you for the question about the lambda parameter and our training mechanism. The lambda parameter, crucial for balancing distillation loss, is analyzed in Figure 7 of Appendix B. We found that MEND performs optimally when lambda is equal to or greater than 1. This balance is key to our finetuning process.
>
>
> **Q3.2. The connection between our two-stage training mechanism and each loss term and “task-specific retraining or compromise in-context learning.**
>
> **A3.2.** Thank you for your query. MEND's two-stage training process is designed to learn a meta-knowledge for
> demonstration distillation that's effective across various tasks without needing retraining for each new task, unlike prompt tuning methods.
> This process helps maintain performance on par with vanilla ICL.
> The first stage aligns the distilled vectors with the LLM's word embeddings, while the second stage fine-tunes this alignment, ensuring adaptability to new tasks.
> This approach differs significantly from methods like HyperTuning, which lack this alignment, resulting in compromised in-context learning performance.
> Our ablation study in Section 5.4 highlights these distinctions and demonstrates MEND's efficacy in preserving in-context learning ability without task-specific retraining.
>
> **(Unfinished)**

---

> ### Author Response · Authors · 2023-11-23
> **Response to Reviewer RPCQ (Part 2)**
>
> **Q4.1. In Figure 3, is the size of the distillation vectors of MEND the same size as PromptTuning?**
>
> **A4.1. Yes, in Figure 3, the size of the distillation vectors for MEND is indeed set to 100, which is consistent with the size used in PromptTuning.**
>
> **Q4.2.  What’s the model size of MEND and FLOPs it introduced (Total parameters and FLOPs)?**
>
> **A4.2.** You can refer to the following table for the size of MEND and its relevant FLOPs for demonstration distillation when compared with PromptTuning.
> It should be noted that the FLOPs include the number of distillations and large language model inference.
>
> | Downstream LLM | Distillation Model | Additional Parameters | Additional TFLOPs |
> |----------------|--------------------|-----------------------|-------------------|
> | gpt2-large     | gpt2               | 137M                  | 0.1               |
> | gpt2-xl        | gpt2               | 137M                  | 0.11              |
> | opt-6.7b       | opt-125m           | 125M                  | 0.11              |
>
> **Q5. In the ablation of pre-training, even if the row is labeled as “No pretraining”, the fine-tuning term still contains the distillation loss. So, what’s the lambda for this row?**
>
> **A5.** We choose the lambda based on the validation performance. The selection of $\lambda$ is shown in the following table:
>
> | Method         | Class->Class | non-nli->nli | non-qa->qa | qa->qa |
> |----------------|--------------|--------------|------------|--------|
> | MEND           | 10           | 1            | 1          | 0.1    |
> | No-Pretraining | 10           | 1            | 10         | 1      |
>
> Distillation loss ($L_{diss}$) also positively contributes to the model’s learning in a no-pretraining setting. This also indicates the importance of imitating the teacher model.

---

### Official Review · Reviewer_Uu3b · 2023-11-01

**Soundness:** 3 good
**Presentation:** 3 good
**Contribution:** 3 good
**Rating:** 8
**Confidence:** 4

**Summary:**

This paper focuses on making in-context learning (ICL) with LLMs more efficient and effective. Vanilla ICL requires one to provide a collection of demonstrations, i.e., task-specific example-label pairs, as context to the LLM while running an inference for a test example. However, this leads to long input sequences, increasing the cost of inference with LLMs where the self-attention cost scales quadratically with input sequence length. This paper proposes to utilize a **demonstration distillation model** to compress the long demonstration sequence into a small number of vectors which can be fed into the LLM as a prompt vector (akin to prompt tuning) during inference. There are existing approaches that adopt such an approach to make ICL efficient. However, those approaches often result in performance degradation. This paper proposes a knowledge distillation-based approach to train the demonstration distillation model. Through extensive empirical evaluations, the paper shows that the resulting demonstration distillation model not only realizes efficient inference by reducing the context length for LLM but often also improves the performance compared to vanilla ICL.

**Strengths:**

1) The paper proposes a novel solution to improve the demonstration distillation approach to make ICL efficient. The solution utilizes knowledge distillation in two stages -- pre-training and task-specific finetuning -- to improve the demonstration distillation model.
2) The paper provides thorough empirical evidence that the proposed approach makes the ICL efficient while also being on par with/improving vanilla ICL.
3) The paper presents a detailed ablation study to highlight the utility of various components of the proposed approach.

**Weaknesses:**

There are no major weaknesses in the paper that the reviewer could find. Please see the questions section below for some clarifying questions.

Some minor comments about improving the quality of presentation are as follows:

1) Consider paraphrasing some sentences to make them clearer:

* On page 2, "Considering the evident a misalignment where the LLM trains on natural language tokens but infers using distillation vectors..."
* In Appendix D, "This will not only lose the information from the discarded tokens and cannot distill demonstration with large $K$ (e.g. $K > 1000$ (Hao et al., 2022))."

2) On page 2, "...we embarked on a in-depth..." --> "...we embarked on **an** in-depth..."
3) In Table 2, "0-shot" --> "zero-shot"
4) In Appendix D, "...are shading insights for future work." -->  "...shedding insights for future work."?

**Questions:**

1) After Eq. (1), the paper states "...where $\mathcal{C}$ is the unique set of $\{y\_i\}\_i=1^K$...". Isn't this a restrictive assumption? There could be tasks where test examples may have a label not present in any of the $K$ demonstrations, e.g., factual QA.
2) In Section 3.2, during the pre-training phase, does one begin with a pre-trained or randomly initialized demonstration distillation model?
3) Do bars in Figure 3 include the cost of generating $S_D$ (distillate vectors) during inference?
4) Why is prompt tuning performance missing from Figure 4a? Similarly, why is vanilla ICL (with truncated demonstration) missing from Figure 4b?
5) On page 8, the paper states ``Moreover, it is noteworthy that performance improves in most cases when the No Input perturbation is applied. This not only **underscores the significance of labels**...`` Could authors expand on this? Why does having only labels in the context help?

---

> ### Author Response · Authors · 2023-11-23
> **Response to Reviewer Uu3b**
>
> Thanks for your recognition of the novelty and usefulness of our work. We are happy to answer your questions as follows.
>
> **Q1. Unclear clarification of   $\{y_i\}_{i=1}^K$ . There could be tasks where test examples may have a label not present in any of the $K$ demonstrations, e.g., factual QA.**
>
> **A1**. Apologies for any confusion caused by our initial explanation. In our paper, $C$ is the unique set of $\{y_i\}_{i=1}^K$ or the set of answer options in question-answering tasks.
>
>
>
> **Q2. In Section 3.2, during the pre-training phase, does one begin with a pre-trained or randomly initialized demonstration distillation model?**
>
> **A2.** In our study, during the pre-training phase of Section 3.2, we start with a pre-trained language model for the two-stage demonstration distillation learning.
>
> **Q3. Do bars in Figure 3 include the cost of generating  (distillate vectors) during inference?**
>
> **A3.** Yes.
>
> **Q4.  Why is prompt tuning performance missing from Figure 4a? Similarly, why is vanilla ICL (with truncated demonstration) missing from Figure 4b?**
>
> **A4.**
> - The exclusion of prompt tuning from Figure 4a is intentional and based on its underlying methodology. Prompt tuning fundamentally differs from the other methods in that it learns soft embeddings from the entire training set rather than utilizing demonstrations from the training dataset as context. Consequently, the volume of demonstrations, which is the focus of Figure 4a, does not impact prompt tuning's performance, making its inclusion in this figure less relevant.
> - In Figure 4b, our focus was on the impact of distilled demonstration length in distillation-based models, which is why vanilla ICL, not using distillation, was initially omitted. Vanilla ICL operates differently, for instance, using only the first token of a demonstration when the distillation vector length is 1, contrasting with our study's emphasis.
>
> **Q5. On page 8, the paper states Moreover, it is noteworthy that performance improves in most cases when the No Input perturbation is applied. This not only underscores the significance of labels... Could authors expand on this? Why does having only labels in the context help?**
>
> **A5.** Thank you for your inquiry. Our observation that performance often improves with the 'No Input' perturbation, where only labels are present in the context, can be attributed to a couple of key factors:
> - No input forces the model to focus solely on the labels, thereby reducing the potential noise or distraction from extraneous or less relevant input data.
> - Labels alone provide a strong signal for the model, enabling it to make more accurate predictions. This could be due to the distilled essence of the task captured in the labels, which the model can leverage more effectively in the absence of other input information. It highlights the model's capacity to extract and utilize the core, task-relevant information from the labels, demonstrating a form of efficiency in its learning process.
>
> These observations suggest that the model can extract and utilize critical, task-relevant information from the labels effectively, demonstrating an efficient learning process. Additionally, recent studies, such as [1], have shown similar improvements with no-input perturbations, further supporting our findings."
>
> [1] Ye, Qinyuan, Iz Beltagy, Matthew E. Peters, Xiang Ren, and Hannaneh Hajishirzi. "FiD-ICL: A Fusion-in-Decoder Approach for Efficient In-Context Learning." In Proceedings of the 61st Annual Meeting of the Association for Computational Linguistics (Volume 1: Long Papers), pp. 8158-8185. 2023.

---

> ### Comment · Reviewer_Uu3b · 2023-12-03
> **Thank you for your response**
>
> Thank you for addressing my questions. After going through the author response and other reviewers' comments, I have decided to keep my score. I believe that additional results on *larger* language models have further strengthened the importance of the proposed method in this work.

---

### Official Review · Reviewer_RDfP · 2023-11-06

**Soundness:** 2 fair
**Presentation:** 3 good
**Contribution:** 3 good
**Rating:** 5
**Confidence:** 5

**Summary:**

This paper proposes MEND: meta-demonstration distillation. The authors design a distillation method to compress a long text demonstration into a short vector. MEND is designed as a meta-distillation method, such that the distillation model can be applied to unseen tasks. Experiments are provided to demonstrate the effectiveness of the proposed method.

**Strengths:**

* The proposed distillation method for prompt-tuning is well-motivated. The presentation is very clear. The proposed distillation method is effective and easy to understand.

* The authors investigate several flavors of models to demonstrate the effectiveness of the proposed method. Specifically, the authors use GPT-2 with different sizes and T5 to show that MEND outperforms existing in context learning approaches.

**Weaknesses:**

My main concern is about experimental settings.

* Could the authors explain why GPT-2 and T5 are used? These models are usually considered outdated and more recent models should be used.
  * For the GPT family, GPT-J, GPT-Neo, OPT are all open-sourced. And the LLaMa models are instruction fine-tuned such that they may show different behavior when facing vectorized demonstrations.
  * For the T5 model, I suggest using Flan-T5, which shows much stronger performance than T5.

* The authors should consider more baselines. For example, Chain-of-Thought (CoT) can demonstrate stronger performance than vanilla ICL. The authors need to at least compare with the vanilla CoT. I also suggest distilling CoT prompts (demonstrations) and see whether this can further improve the performance of MEND.

I will raise the score if the authors can run some experiments on more recent models.

**Questions:**

See above

---

> ### Author Response · Authors · 2023-11-23
> **Response to Reviewer RDfP**
>
> Thanks for your detailed and constructive review. We would like to address your concerns as follows.
>
> **Q1.1. Could the authors explain why GPT-2 and T5 are used? These models are usually considered outdated and more recent models should be used.**
>
> **A1.1.** Thank you for your question regarding our choice of GPT-2 and T5 as the backbone language models. We selected these models based on their demonstrated capabilities in in-context learning, as highlighted in related works [1-2]. Despite being older models, GPT-2 and T5 are still widely recognized for their effectiveness in this domain.
>
> Moreover, these models embody different architectural styles, with GPT-2 being a decoder-only model and T5 being an encoder-decoder model. This diversity allows us to showcase the general applicability of our proposed method across varying model structures. By achieving positive results with both GPT-2 and T5, we demonstrate that our method is not limited to a specific architecture but is broadly applicable, which is crucial for validating the robustness and versatility of our approach in the context of in-context learning.
>
> [1] Min, Sewon, Mike Lewis, Luke Zettlemoyer, and Hannaneh Hajishirzi. "Metaicl: Learning to learn in context." arXiv preprint arXiv:2110.15943 (2021).
> [2] Phang, Jason, Yi Mao, Pengcheng He, and Weizhu Chen. "Hypertuning: Toward adapting large language models without back-propagation." In International Conference on Machine Learning, pp. 27854-27875. PMLR, 2023.
>
>
>
> **Q1.2. More recent language model should be used.**
>
> **A1.2.** Thanks for the suggestion. We have added an experiment that tests our method on opt-6.7b and flan-t5-xl. You can refer to the general response for more information.
>
> **Q2. The authors should consider more baselines. For example, Chain-of-Thought (CoT) can demonstrate stronger performance than vanilla ICL. The authors need to at least compare with the vanilla CoT. I also suggest distilling CoT prompts (demonstrations) and see whether this can further improve the performance of MEND.**
>
> **A2.** Thank you for the suggestion to include Chain-of-Thought (CoT) baselines.
> Our current dataset doesn't have an ID match with existing CoT sources, which limited our ability to directly compare with standard CoT prompts.
> However, we've addressed this by utilizing the BIG-Bench-Hard (BBH) dataset, a well-known CoT evaluation benchmark.
> We adapted BBH problems into multiple-class classification tasks and evaluated the CoT capabilities of MEND and other baselines like HyperTuning. The results, as shown in the table below, demonstrate the effectiveness of our distillation model in capturing meaningful CoT for the downstream large language model.
>
> Notably, MEND shows superior performance compared to the baseline methods, indicating its enhanced ability to distill Chains for effective language model reasoning.
>
> | Methods     | BBH Accuracy |
> |-------------|--------------|
> | Vanilla ICL | 0.3243       |
> | HyperTuning | 0.3503       |
> | MEND        | **0.3558**   |

---

### Official Review · Reviewer_Dw9E · 2023-11-10

**Soundness:** 3 good
**Presentation:** 2 fair
**Contribution:** 3 good
**Rating:** 6
**Confidence:** 4

**Summary:**

This work proposes a new method for efficient in-context learning via direct prediction of demonstration context vectors. The proposed method, called MEND, combines hypernetwork training with distillation of regular in-context learning behavior to achieve high-quality "prompt vector" synthesis capabilities. Authors validate MEND on the MetaICL dataset using GPT2 and T5 models, showing performance gains and accuracy improvements compared to other in-context learning baselines.

---

Post-rebuttal update: I thank the authors for their response and clarifications, and I am keeping my current score.

**Strengths:**

* The proposed approach is well-motivated and achieves significant improvements in each of the studied setups.
* The paper contains a detailed analysis section along with the ablation study for MEND, justifying the necessity of each component of the method.

**Weaknesses:**

* My primary concern regarding the evaluation is that the models studied in the paper (GPT2-XL, T5-large) are relatively small and not representative of models that actually benefit from in-context learning. In fact, authors acknowledge this limitation in the appendix; I simply believe that having experiments on larger models (for example, training only one distillation model and applying it to larger LMs) would increase the impact of the work.
* The inference efficiency measurement protocol could likely be improved. First, it is unclear whether key/value caching is used for generation: this should make the impact of additional demonstrations less severe. Also, I think it would be helpful to have a more detailed memory/time breakdown for MEND: measuring only the inference with obtained meta-demonstrations is not sufficient, as the distillation model needs to process input demonstrations into prompts.
* At times, it was a bit difficult to understand the reasoning of the paper due to grammar errors/typos and word choice. Consider, for examplem, "the evident an misalignment" and "between teacher student's" on page 2, "into a condensed vectors", "distill the supervisional from demonstrations", and "cannot leveraging" on page 5.

**Questions:**

* In Table 2, what were the standard deviations across runs?
* How did you format 16 demonstration examples into a single input string for each dataset? For example, there are different ways of joining several demonstrations (space, linebreak etc.). Have you studied the robustness of models/methods to that formatting?

---

> ### Author Response · Authors · 2023-11-23
> **Reponse to Reviewer Dw9e (Part 1)**
>
> Thanks for your valuable time and thoughtful review. We want to resolve your concerns as follows:
>
> **Q1. Other large language model. **
>
> **A1.** Thanks for the suggestion. We have evaluated the generazation of our proposed method on flan-t5-xl and opt-6.7b. You can refer to the general response for more information.
>
> **Q2.1. The inference efficiency measurement protocol could be improved. First, it is still being determined whether key/value caching is used for generation: this should make the impact of additional demonstrations less severe.**
>
> **A2.1** Our experiments did not employ key-value caching across all baseline methods, including MEND, to ensure a consistent and fair comparison framework. Our observation also influenced our decision that key-value caching does not significantly enhance Vanilla ICL's efficiency in the context of our study in all the evaluation metrics. In our efficiency evaluation experimnent on opt-6.7b, we can observe key-value cache reduce the FLOPS but still consume large memory and require much computation time.
>
> |      Method     | TFLOPS | GPU Memory (GB) | Time (Seconds) |
> |:---------------:|--------|-----------------|----------------|
> |   Vanilla ICL   | 19.06  | 2.95            | 1.44           |
> |   PromptTuning  | 4.93   | 0.70            | 0.70           |
> |       MEND      | 5.03   | 0.70            | 0.74           |
> | Key-Value Cache | 0.93   | 2.85            | 2.43           |
>
> Another key consideration was the potential loss of contextual information from the demonstration interactions when using key-value caching. This loss could necessitate additional meta-training for the language model to effectively represent demonstration examples, which falls outside the scope of our current investigation [1].
>
> [1]Qinyuan Ye, Iz Beltagy, Matthew Peters, Xiang Ren, and Hannaneh Hajishirzi. 2023. FiD-ICL: A Fusion-in-Decoder Approach for Efficient In-Context Learning. In Proceedings of the 61st Annual Meeting of the Association for Computational Linguistics (Volume 1: Long Papers), pages 8158–8185, Toronto, Canada. Association for Computational Linguistics.
>
> **Q2.2. Also, it would be helpful to have a more detailed memory/time breakdown for MEND: measuring only the inference with obtained meta-demonstrations is not sufficient, as the distillation model needs to process input demonstrations into prompts.**
>
> **A2.2.**  When evaluating MEND, we calculated the total time inclusive of both obtaining the demonstration distillation and utilizing the distillation for LLM inference. This approach was chosen to provide a comprehensive understanding of the efficiency of our method in real-world scenarios. If the meta-demonstrations were pre-obtained, the efficiency would align closely with that of prompt tuning. However, our focus was to assess the end-to-end efficiency of MEND, reflecting its practical application where demonstrations may not always be pre-processed.
>
> **Q3. Readability improvement.**
> **A3.** Thanks for your suggestion; we have updated our paper based on your comments.
>
> **Q4. In Table 2, what were the standard deviations across runs?**
> **A4.** Thanks for the suggestion.  We have uploaded the standard deviation for Table 2. You can refer to our updated paper.
>
> **(Unfinished)**

---

> > ### Author Response · Authors · 2023-11-23
> > **Reponse to Reviewer Dw9e (Part 2)**
> >
> > **Q5. How did you format 16 demonstration examples into a single input string for each dataset? For example, there are different ways of joining several demonstrations (space, linebreak, etc.). Have you studied the robustness of models/methods to that formatting?**
> > **A5.**
> > - _Template Selection_ : In our experiments, we formatted the 16 demonstration examples into a single input string using whitespace to concatenate the input and output within each demonstration and different demonstrations. This choice was inspired by the related work MetaICL[1], to meet the need for a simple and consistent separator that could be easily replicated in various settings.
> > - _Robustness towards Different Templates_: To test the robustness of MEND against different formatting styles, we also conducted experiments using new lines to separate the input and output and two new lines to separate different demonstrations.
> > For methods that require meta-training(HyperTuning and MEND), we trained them on datasets using whitespace as a separator and evaluated on datasets separated by newlines. The consistent performance across these different formats further underscores the adaptability of MEND.
> > Notably, MEND exhibited impressive robustness to these formatting differences, showing a minimal performance variation of less than 0.3% between using spaces and newlines.
> >
> > | Methods     | whitespace | newline | Diff. |
> > |-------------|------------|---------|-------|
> > | Vanilla ICL | 41.30      | 38.90   | -2.40 |
> > | HyperTuning | 40.42      | 40.08   | -0.34 |
> > | MEND        | 43.35      | 43.50   | +0.15 |
> >
> > This indicates that MEND's performance is not significantly impacted by the choice of separator, which is an important consideration for its practical application.
> >
> > [1] Min, Sewon, Mike Lewis, Luke Zettlemoyer, and Hannaneh Hajishirzi. "Metaicl: Learning to learn in context." arXiv preprint arXiv:2110.15943 (2021).

---

### Author Response · Authors · 2023-11-23
**General Response**

**Q1. Experiments on other large language models.**

**A1.** Thank you for your insightful feedback regarding the scalability of our proposed method, MEND, to larger language models. Recognizing the importance of this aspect, we have conducted additional experiments with larger models such as flan-t5-xl (3B) and opt-6.7b. The results from these experiments are encouraging – MEND not only consistently matched but, in several cases, surpassed Vanilla ICL in terms of performance.
Furthermore, the efficiency gains with these larger models were significant. For instance, we noted a reduced computational overhead by 70% FLOPs in opt-6.7b, which underscores MEND's potential for practical, large-scale deployments. This demonstrates not only MEND's adaptability to larger models but also its ability to maintain and, in some instances, enhance performance while substantially reducing computational demands.
We acknowledge that exploring MEND's application to even larger models represents a valuable direction for future research and are committed to further investigating this avenue. Again, thank you for emphasizing this point, which significantly contributes to the robustness and applicability of our work.
For more details about this discussion, you can refer to Appendix D.

| Backbone Model | Class->Class |
|:--------------:|:------------:|
|  **opt-6.7b**  |              |
|   Vanilla ICL  |     42.38    |
|  PromptTuning  |     38.81    |
|   HyperTuning  |     42.67    |
|      MEND      |     44.27    |
| **flan-t5-xl** |              |
| Vanilla ICL    | 40.63        |
| PromptTuning   | 33.24        |
| HyperTuning    | 39.70        |
| MEND           | 40.70        |

Inference efficiency of opt-6.7b on Class->Class task.
|      Method     | TFLOPS | GPU Memory (GB) | Time (Seconds) |
|:---------------:|--------|-----------------|----------------|
|   Vanilla ICL   | 19.06  | 2.95            | 1.44           |
|   PromptTuning  | 4.93   | 0.70            | 0.70           |
|       MEND      | 5.03   | 0.70            | 0.74           |

---

### Meta-Review · Area_Chair_K7k3 · 2023-12-05

**Metareview:**

This paper introduces MEND, a novel method for efficient in-context learning through meta-demonstration distillation. Reviewers appreciate the clear presentation and the method's motivation, showing improvements over existing in-context learning baselines. However, concerns were raised about the evaluation using relatively smaller models like GPT-2 and T5, suggesting the need for experiments on larger, more contemporary language models. Suggestions were made to enhance the inference efficiency measurement protocol and improve the paper's readability. Reviewers unanimously acknowledge the thorough empirical evidence and the detailed ablation study supporting the proposed approach's efficacy. While some minor clarity issues were noted, the authors' response post-rebuttal seems to address concerns, particularly by providing additional results on larger language models, reinforcing the significance of their method in the field of in-context learning. Overall, the paper's contributions remain substantial, given the addressed concerns post-rebuttal.

**Justification For Why Not Higher Score:**

While the paper's contributions were acknowledged, there were suggestions to provide more comprehensive explanations regarding how the proposed method maintains or enhances in-context learning abilities. Incorporating these suggestions could strengthen the paper and potentially elevate its impact.

**Justification For Why Not Lower Score:**

The paper's core contributions, clarity in presentation, and the novelty of the proposed method for efficient in-context learning through meta-demonstration distillation were recognized positively by the reviewers.

---

### Decision · Program_Chairs · 2024-01-16

Accept (poster)